# Insight into Hyper-Branched Aluminum Phosphonate in Combination with Multiple Phosphorus Synergies for Fire-Safe Epoxy Resin Composites

**DOI:** 10.3390/polym12010064

**Published:** 2020-01-01

**Authors:** Yao Yuan, Bin Yu, Yongqian Shi, Long Mao, Jianda Xie, Haifeng Pan, Yuejun Liu, Wei Wang

**Affiliations:** 1Fujian Provincial Key Laboratory of Functional Materials and Applications, School of Materials Science and Engineering, Xiamen University of Technology, Xiamen 361024, China; yuanyao@mail.ustc.edu.cn (Y.Y.); maolong0412@163.com (L.M.); xiejianda@xmut.edu.cn (J.X.); 2Centre for Future Materials, University of Southern Queensland, Toowoomba, QLD 4350, Australia; ahu07yb@gmail.com; 3College of Environment and Resources, Fuzhou University, Fuzhou 350002, China; shiyq1986@fzu.edu.cn; 4Faculty of Engineering, China University of Geosciences (Wuhan), Wuhan 430074, Hubei, China; hfpan19@163.com; 5State Key Laboratory of Fire Science, University of Science and Technology of China, Hefei 230026, China

**Keywords:** epoxy resin, hyper-branched flame-retardant, smoke suppression, mechanism

## Abstract

Epoxy resin (EP) has widespread applications in thermosetting materials with great versatility and desirable properties such as high electrical resistivity and satisfactory mechanical properties. At present, 9,10-Dihydro-9-oxa-10-phosphaphenanthrene-10-oxide (DOPO) is widely applied to EP matrix for high flame resistance. Nevertheless, EP/DOPO composites acquire highly toxic decomposition products and smoke particles produced during combustion due to the gaseous fire-inhibition mechanism, which will be a major problem. To address this concern, an effective hyper-branched aluminum phosphonate (AHPP) was rationally designed and then coupled with DOPO into EP matrix to fabricate the fire-safe epoxy resin composites. On the basis of the results, significant increment in limiting oxygen index value (an achievement of 32% from 23.5% for pristine EP) and reduction in peak heat release rate and total heat release (59.4% and 45.6%) with the DOPO/AHPP ratio of 2:1 were recorded. During the cone calorimeter test, both the smoke production and total CO yield of EP-4 composite with the DOPO/AHPP ratio of 1:2 were dramatically decreased by 42.7% and 53.6%, which was mainly associated with the excellent catalytic carbonization of AHPP submicro-particles for EP composite. Future applications of submicro-scaled flame-retardant with various phosphorus oxidation states will have good prospects for development.

## 1. Introduction

Epoxy resin (EP) belongs to an outstanding class of thermosetting polymers, which has been considered to be a widely used material owing to its multiple and unique properties, for instance low shrinkage, superior solvent resistance, remarkable electrical insulation, excellent adhesive strength and ease of curing and processing [1,2,3,4]. Nonetheless, as an organic polymer material, EP produces toxic fumes (especially CO) and soot particles (smoke) during the combustion process, which greatly damages our natural environment and seriously restricts its potential applications in aerospace, coating, and electrical device fields [5,6,7,8]. Accordingly, under safety consideration, incorporating appropriate synergists to fabricate the fire-safe epoxy resin composites is an urgent and challenging issue for fire safety design [9,10,11].

In the past few years, different approaches have been proposed by introducing various halogen-free flame retardants containing phosphorus, nitrogen, and silicon into the EP composites without environmental issues [12,13,14]. Generally, phosphorus-containing flame-retardant is considered to be the most efficient method among the above systems [15,16]. Notably, 9,10-Dihydro-9-oxa-10-phosphaphenanthrene-10-oxide (DOPO) has been commercially applied in various polymer materials, especially EPs [17,18]. The DOPO structure was covalently incorporated into the backbones of EPs by the reaction between P–H bond and epoxy groups, which greatly increases the dispersion and durability of the flame-retardant. Unfortunately, highly toxic decomposition products and soot particles generated under fire conditions are the critical defects of the flame-retardant. Qiu et al. [19] demonstrated that the simultaneous existence of TDBA (phosphaphenanthrene derivative) and DOPO structure in the EP composites produced more pyrolysis products and formed large-scale smoke particles, signifying the prominent gas phase action of DOPO present in inhibiting the transformation of the matrix to fuel.

On the basis of previous investigation, the chemical environment of phosphorus affects its flame-retardant efficiency. Braun et al. [20] compared the effect of phosphorus valence on the fire behavior of EP and found that the flame inhibition effect, i.e., gas phase action decreased with the increasing oxidation state. Generally, flame retardants with a higher oxidation state show only condensed phase action compared to the lower oxidation state, which mainly functions in the gas phase [21]. Unfortunately, there are still no reports on the phosphorus-containing flame-retardant EP composites with the combined effects of various oxidation states. Furthermore, a large amount of research on phosphorus-based additives with various phosphorus-containing groups in different valence should be attractive for advanced application.

Potentially owning phosphorus-rich groups, hyper-branched polymers possess interesting architecture and a large number of active terminal groups compared to their linear counterparts [22,23], and have attracted increasing attention in constructing highly-efficient flame retardants. Traditionally, hyper-branched polymers are prepared mainly by polymerization of A_2_ + B_3_, and have received a lot of attention for the unusual architecture and physicochemical properties [24]. Meanwhile, aluminum salts have attracted great interest recently due to their practical advantages such as facile preparation, being easy to handle, none of the toxic gas, and no health safety and environmental impacts. Yang et al. [25,26] incorporated aluminum hypophosphite (AHP), melamine, and polycarbonate into reinforced poly(1,4-butylene terephthalate), showing excellent flame retardance and reaching UL 94 V-0 criterion, while in neither study was the smoke hazard of PBT composites taken into consideration. Pinto et al. [26] prepared flame-retardant thermoplastic polyurethane elastomer composites using aluminum hydroxide (ATH), which offers the advantage of acting as flame-retardant as well as a smoke suppressant. However, 70 wt. % loading is still too high.

To obtain EP composites with simultaneous excellent flame retardance and smoke hazard suppression, a novel and submicro-scaled hyper-branched aluminum phosphonate (AHPP) was successfully prepared. The chemical structures of the precursor and the final product were confirmed by a great deal of structural characterization and morphologic assessments. This work extends a new path for providing the synergistic effect between AHPP and DOPO with various oxidation states (–1, +1, +3) on improving smoke suppression and fire safety for polymeric materials during combustion. Furthermore, the flame retardation and smoke suppression mechanisms were also systematically studied through various measurements. The detailed characterization information is supplied in the Appendix A.

## 2. Materials and Methods

### 2.1. Raw Materials

Diglycidyl ether of bisphenol A (DGEBA, E-44, epoxy value = 0.44 mol/100 g) was purchased from Jiangfeng Chemical Industry Co. Ltd. (Hefei, China). Benzene phosphorus oxydichloride (BPOD) and DOPO were obtained from Sun Chemical Technology Co., Ltd. (Shanghai, China). 4, 4′-diaminodiphenylmethane (DDM), hydrogen peroxide (30% aq.), barium hydroxide (analytical reagent grade), acetone, aluminum sulfate (Al_2_(SO_4_)_3_·18H_2_O), and Tetrakis(hydroxymethyl)phosphonium sulfate (THPS, technical pure grade) were purchased from Sinopharm Chemical Reagent Co. Ltd. (Shanghai, China).

### 2.2. Synthesis of the Precursor

Tris(hydroxymethyl)phosphine oxide (THPO) was prepared according to our previous work [27] and the preparation process of the precursor was successfully prepared (see Figure 1). All of the raw materials are nontoxic and environment-friendly. In a typical experiment, a three-necked flask was charged with 0.3 mol of BPOD under stirring, followed by slowly adding 0.15 mol of THPO to the suspension. The polymerization was carried out at 100 °C for 4 h with constant agitation under nitrogen. A white solid with a ca. 98% yield was obtained after the purification of the hyper-branched phosphonate (HPP).

### 2.3. Synthesis of AHPP Submicro-Particles

The precursor was dissolved in 50 mL H_2_O under stirring in a three-necked flask. A standardized sodium hydroxide solution (0.1 M) was added, grown at pH 7.5–8. While the mixture turned colorless sticky liquid, 0.05 mol of aluminum sulfate was dissolved in 20 mL H_2_O and dripped into the reaction system at 70 °C for 20 min. After cooling to room temperature within another 30 min, more white granules appeared. Successively, the resulting products, i.e., hyper-branched aluminum phosphonate (AHPP), were washed and filtrated with water several times. The final products were dried at 100 °C and the yields were >90.0%.

### 2.4. Fabrication of EP/DOPO-AHPP Composites

AHPP was firstly mixed into DGEBA and dispersed in acetone, followed by adequate mechanical stirring at 90 °C for 1 h. DDM was poured into the mixture after removing the solvent. Subsequently, it was added to the preheated mold, curing at 100 °C for 2 h and post-curing at 150 °C for another 2 h. After cooling, the EP composites with AHPP and DOPO were labeled as EP/AHPP/DOPO in various proportions using this similar procedure and the formulations of EP samples are presented in Table 1.

## 3. Results and Discussion

### 3.1. Structural Characterizations

Submicro-scaled hyper-branched aluminum phosphonate (AHPP) was synthesized via the dehydrochlorination between –OH in THPO and O=P–Cl in BPOD. The chemical structure of the precursor was determined by the NMR spectrum at first. Appendix A presents the multiplets in the ranges of 7.57–8.30 ppm and 7.39–7.51 ppm, corresponding to aromatic rings. The signal from 10.45 ppm is characteristic of the hydroxy proton of O=P–OH and the peak at 5.5 ppm is ascribed to the hydroxy proton of P–CH_2_–OH. Additionally, the chemical shift at 3.99 to 4.24 ppm are characteristic of the methylene proton of Ph-P(=O)–O–CH_2_-P=O. As depicted in Appendix A, the signal at 16.8 ppm is associated with the phosphorus of O–(Ph)P(=O)–O, and the peak at 13.7 ppm is attributed to the phosphorus of O–P=O(Ph)–O. Three peaks located at 43.2, 40.4, and 37.4 ppm are due to the phosphorus of O=P–(CH_2_–)_3_ with one, two, and three hydroxyls reacted, respectively, which are designated as terminal (T), linear (L), and dendritic (D) units [28]. These results confirm that the precursor was synthesized successfully.

The structure of the target product is further confirmed by FTIR spectroscopy. As shown in Figure 2b, the emergence of absorption peaks is characteristic of several function groups, such as O–H (3415 cm^−1^), C–H (2920, 2850 cm^−1^), P–Ph (1440 cm^−1^), and P–CH_2_ (690, 750 cm^−1^) [29]. The signal from 1150 cm^−1^ is attributed to the P=O and O=P–C stretching modes. Furthermore, the stretching vibrations at 1635, 1542, and 1494 cm^−1^ are characteristic of the skeletal vibration in aromatic rings [30]. Notably, the peak at 1080 cm^−1^, which is ascribed to the P–O–C bond in AHPP, confirms the successful linkage between BPOD and THPO [31]. 

The surface morphology of the AHPP submicro-particles was observed by SEM and TEM, as shown in Appendix A. The fact that the AHPP particles are well-dispersed and uniform without agglomeration is remarkable. Moreover, EDX was also used to determine the surface elements of AHPP particles. The characteristic peaks of P, O, and Al have been detected in the EDX spectrum, further confirming the successful preparation.

### 3.2. Characterization of the Degradation Behavior

The thermal stability of pristine EP and its composites are depicted in Figure 2, and the corresponding thermograms of the resultant materials are summarized in Table 2. From the DTG curves, the cured epoxy and its composites display a single-step process by anaerobe decomposition (Figure 2d). Under nitrogen atmosphere, EP/DOPO or EP-AHPP induce lower *T_d_* value (EP: 367 °C, EP/DOPO: 279 °C, EP-AHPP: 329 °C) and can be attributed primarily to the fact that the O=P–O and P–O–C bonds are more unstable than the normal C–C bond [32]. By contrast, AHPP increases the number of char residues than DOPO does. Additionally, when AHPP and DOPO are introduced simultaneously into EP matrix in certain content, the EP/DOPO-AHPP system sustains initial decomposition temperature at high temperature and increases the volume of char residues at 800 °C. Ultimately, from the DTG curves, it is clear that significant improvement in thermal stability can be achieved by the simultaneous incorporation of AHPP and DOPO with higher char residue during combustion. 

Basically, RT-FTIR analysis provides useful information about the chemical structure changes to ascertain the degradation behavior of pristine EP, EP/DOPO, EP-AHPP, and EP/AHPP samples. The main peaks and bands of EP and its composites are shown in Figure 3a,b and Appendix A. As for the marked regions, it is very clear that the peaks at 3510, 2930, 2860, 1603, 1502, 1360, 1110, and 820 cm^−1^ belong to EP [33]. The intensity of the band at 2860 and 2930 cm^−1^ resulting from symmetric and CH_2_ asymmetric vibrations disappears completely at 550 °C, indicating the main chain of the epoxy has been completely degraded during the heating process [34]. With the temperature increasing, only the absorption peaks at 1603, 1502, 1110, and 820 cm^−1^ are maintained, which could be attributed to the char residue with a multi-aromatic structure formed [35]. As a result, the P–O–P, P–O–C, and O=P–O– vibrations at 1250 and 1090 cm^−1^ for EP-2 and EP-4 are found in the FTIR spectra [36], signifying that the presence of AHPP or DOPO-AHPP results in the formation of protective residues during burning [37]. Accordingly, the result clearly indicates that the formation of char residue for the EP/DOPO-AHPP composites performs strong physical barriers during combustion to increase the fire safety of EP composites.

### 3.3. Flame Retardance

The effect of phosphorus-containing flame retardants with various phosphorus oxidation states on the ignitability of pristine epoxy and its composites were investigated by LOI and vertical burning UL-94 test. As shown in Table 3, pristine EP has intrinsic flammability with LOI value of 23.5% and no classification during UL-94 tests. In comparison with the EP/DOPO or EP-AHPP sample, the LOI value of EP/DOPO-AHPP significantly increases. Specially, when the ratio of DOPO and AHPP is 2:1, the LOI value rises to 32%. Moreover, the cured EP/DOPO-AHPP passes the UL-94 V-0 rating without dripping, indicating that the incorporation of AHPP and DOPO can obtain an enormous improvement in flame retardance of EP composites. Obviously, the EP-3 sample has excellent flame retardance, which may further promote the potential application value of EP/DOPO-AHPP with various phosphorus valences.

Successively, to get further information about the combustion behavior, the combustion process of epoxy thermoset at oxygen concentrations of 23.5% and 32% during LOI test was monitored. As depicted in Figure 3c, pristine EP ignites rapidly and burns vigorously over the surface of the thermoset. Contrarily, the incorporation of AHPP and DOPO changes the combustion behavior of EP matrix significantly. EP/DOPO-AHPP exhibited a so-called “blowing-out effect”, reflecting in the raised LOI values from 23.5% up to 32.0%. Besides, EP/DOPO-AHPP forms expandable burning residues during combustion. It is apparent that the flame-retardant system promotes the production of a physical barrier layer, retarding the permeation of heat and oxygen [38]. Accordingly, a favorable synergistic effect for the fire safety of EP/DOPO-AHPP composite is revealed and has been further confirmed. 

Cone calorimeter measurement is experimented to reveal the flammability of the polymeric materials in bench-scale measurements [39]. As portrayed in Figure 4a,b and Table 3, pristine epoxy is highly flammable with a peak heat release rate (PHRR) of 1422 kW/m^2^. Meanwhile, it is noticeable that several sharp peaks are observed at 100–200 s for pristine epoxy (Figure 4a), which is attributed to the formation of a protective chars [40]. When 5 wt. % AHPP or DOPO is incorporated separately, the PHRR values are decreased dramatically to 1015 and 832 kW/m^2^, an approximately 28.6% and 41.4% reduction in contrast to pristine EP. A larger reduction (59%) of PHRR is seen with the simultaneous presence of AHPP and DOPO, exhibiting the potential to achieve better flame retardance. At the end of combustion, pristine epoxy burns fiercely with a total heat value of 111.9 MJ/m^2^ (Figure 4b) and loses ~95% of its initial mass. As Figure 4 demonstrates, the addition of 5 wt. % AHPP or DOPO decreases the THR to 69.4 MJ/m^2^ (EP-AHPP) and 72.6 MJ/m^2^ (EP/DOPO). It should be highlighted that the simultaneous incorporation of AHPP and DOPO can obviously reduce both PHRR and THR of EP composites when the ratio of DOPO/AHPP is 2:1. Compared with the corresponding samples, the results show that the synergy between AHPP and DOPO reduces the heat release for the epoxy matrix during combustion, which is beneficial for decreasing the thermal hazard.

### 3.4. Smoke and Toxicity Hazards Analysis

Generally, epoxy resin, an organic polymer with an aliphatic and aromatic ring, produces a large amount of toxic fumes (especially CO) and smoke production. Accordingly, to meet the demands of reducing the death rate in fire and environmental protection, intensive attention should be focused on eco-friendly flame-retardant additives. Here, Figure 4 gives the combustion parameters and smoke toxicity information of EP composites. Compared to pristine EP, the CO release and smoke production escaping from the combustion of EP composites have been suppressed by the simultaneous addition of AHPP and DOPO at 5 wt. % loading (>40% reduction for total smoke production (TSP), Figure 4d in this research. In detail, the existence of DOPO actually increases the smoke production rate (SPR) and CO release. Meanwhile, the trend of higher total smoke production (TSP) is obtained for EP/DOPO, signifying that DOPO has a stronger quenching effect in the gaseous phase. The gaseous phase flame-retardant effect leads to incomplete combustion, which implies that the introduction of DOPO forces more pyrolysis products of polymer into large-scale smoke particles [19]. In contrast, the increase of residual char for the EP-AHPP sample can suppress the transformation of the matrix to fuel, thus reducing smoke production. The elevated production rates of CO for EP/DOPO signify the enhanced incomplete combustion, and the decreased value of the EP-AHPP composite further represents the stronger suppression of smoke toxicity on the combustion reaction. Especially for EP/DOPO-AHPP, the surprisingly lower smoke toxicity and flame inhibition with the same flame-retardant content testify the distinguished condensed phase and gaseous phase flame-retardant activity on decreasing the fire hazard of cured EP during burning. 

To further evaluate the fire safety properties, TG-FTIR measurement was employed to detect the volatile degradation products resulting from EP composites and the absorbance of total pyrolytic gases of EP and EP/DOPO-AHPP composite is displayed in Figure 5. Similar characteristic peaks are clearly distinguished by typical strong FTIR signals: hydroxide groups (3650–3400 cm^−1^), the vibration absorbance of C=O and C–O–C groups (1607, 1260 cm^−1^), and compounds containing aromatic ring (1605–1450, 1260–927 cm^−1^). It shows that the stretching vibrations at 2360 and 2180 cm^−1^ derived from the formation of CO_2_ and CO during decomposition for the epoxy matrix significantly decreases, revealing reduced toxic gases release for EP/DOPO-AHPP. 

To better clarify the smoke toxicity suppression behavior, the FTIR absorbance of volatile degradation products of EP and the EP/DOPO-AHPP composite is displayed in Figure 5c–f. In comparison with the pristine EP, the absorbance intensity of pyrolytic products of the EP/DOPO-AHPP system is dramatically decreased, which will be aggregated to form smoke [41]. Meanwhile, the reduced CO release and the organic volatiles have been significantly decreased, which further confirms the suppression of smoke and toxicity. According to the above-mentioned facts, a fire hazard suppression mode, with the incorporation of AHPP and DOPO, simultaneously reduces the thermal stability and improve the fire safety for EP composite.

### 3.5. Flame Retardation Mechanism

The char analysis of EP and its composites from cone tests provide useful feedback about the flame retardation mechanism, including char morphology and structure which are studied in-depth. As expected, pristine EP exhibits a high flammability with visible surface cracks after combustion (Appendix A). Simultaneously, the presence of AHPP and DOPO can catalyze resin to largely form more stable char. Therefore, the residual char of EP/DOPO-AHPP is much more compact and continuous with fewer cracks and holes than those of other EP composites. 

For EP/DOPO-AHPP, incorporating reactive organic and unreactive inorganic additives leads to the construction of a very thick and non-inflammable char layer after burning, due to the catalytic carbonization of flame retardants [42]. As shown in Appendix A, the elemental mapping image shows that O, P and Al elements (i.e., the residual char constituents, apart from C) are homogeneously distributed. Meanwhile, as can be seen in XRD patterns of the char residue, AHPP particles undergo thermal degradation producing Al_2_O_3_ [43] on the epoxy surfaces, which acts as a protective layer as a heat and oxygen shield to reduce the flammable volatiles escaped and improve the fire safety.

To further evaluate the influence of EP/DOPO-AHPP on the char formation during decomposition, the macroscopic view and graphitic structure of the char residues were studied directly by a digital camera and Raman spectroscopy. As portrayed in Figure 6, adding AHPP and DOPO into EP increases the char yield and strength. As is well-known, flame-retardant efficiency is not only dependent on the number of char residues but also their quality [44]. The spectra of char residue exhibit a similar shape including two peaks at 1590 and 1360 cm^−1^. The peak at 1360 cm^−1^ is named as D band and the peak at 1590 cm^−1^ belongs to G band [45]. Generally, the graphitization degree of char could be calculated by the integrated intensity ratio of D and G bands [46]. The *I*_D_/*I*_G_ value follows the sequence of pristine EP (2.81) > EP/DOPO (2.60) > EP-AHPP (2.58) > EP/DOPO-AHPP (2.57). Therefore, adding AHPP and DOPO could accelerate amorphous char into the graphitic structure during burning, achieving an ideal physical barrier of the EP composites.

From Figure 7, the peaks in the range of 1550–750 cm^−1^ which can be assigned to O=P–O–, P–O–C, and P–O–P vibrations in the FTIR spectra [36]. Additionally, the spectra exhibit a peak at 817 cm^−1^ distinctly, signifying that the presence of AHPP and DOPO can promote resin to form the aromatic structure and phosphorus-rich char during decomposition.

Based on the analysis above, for EP/DOPO-AHPP ternary, the PHRR, THR, and both TSP and toxic CO are considerably decreased. The integrity of char residue is increased, and the phosphorus-containing moieties remain in the condensed phase, suggesting that combustible volatiles release and smoke toxicity is suppressed distinctly. As shown in Figure 8, the generation of residual char with Al_2_O_3_ layer covering on the epoxy surfaces performs a stronger barrier action to protect the EP matrix with the simultaneous addition of AHPP and DOPO. In the gaseous phase, the notable phenomenon is the generation of a so-called “blowing-out effect”, which is consistent with the above-mentioned combustion process and in accordance with previous reports [33,47]. It is noteworthy that the enormous improvement in fire safety for the EP/DOPO-AHPP composites, derived from LOI, UL-94, and cone calorimeter results, are probably connected with the continuous and compact residues, which can avoid oxygen from feeding the fire and inhibit the release of flammable gas, thus resulting in the relatively low THR and TSP of EP/DOPO-AHPP [48]. Therefore, the condensed mechanism and gaseous mechanism for flame retardance and smoke toxicity suppression between AHPP and DOPO are verified, and the former one is dominant in EP/DOPO-AHPP composites.

## 4. Conclusions

A fire hazard suppression mode containing phosphorus oxidation states of −1, +1, and +3 through the incorporation of reactive organic and unreactive inorganic additives was successfully applied to EP, resulting in the construction of a very thick and non-inflammable char layer and tremendous reduction of TSP and toxic CO during combustion. Novel flame-retardant particles were successfully prepared via polycondensation. The submicro-scaled structure of this phosphorus-containing flame-retardant endows resultant EP with excellent smoke hazard suppression property. The incorporation of AHPP and DOPO into EP catalyzed and accelerated the formation of protective barrier char layers, compared with the thermosets filled with AHPP or DOPO individually at a same mass fraction. Notably, with a much higher isolating effect on combustible volatiles, oxygen, and heat than independently produced chemical chars in this fire hazard suppression mode, the EP/DOPO-AHPP system displayed dramatically higher flame retardance and smoke suppression properties, including LOI, UL-94 rating, and cone calorimeter results. All the results showed that the sample EP-3 with the DOPO/AHPP ratio of 2:1 endowed the EP composites with the most satisfying flame-retardant performance, and the later one with the ratio of 1:2 exhibited a better efficiency of smoke toxicity suppression. As a result, this fire hazard suppression mode exhibits an efficient approach for enhancing the fire safety of thermoset polymers with high performance.

## Figures and Tables

**Figure 1 polymers-12-00064-f001:**
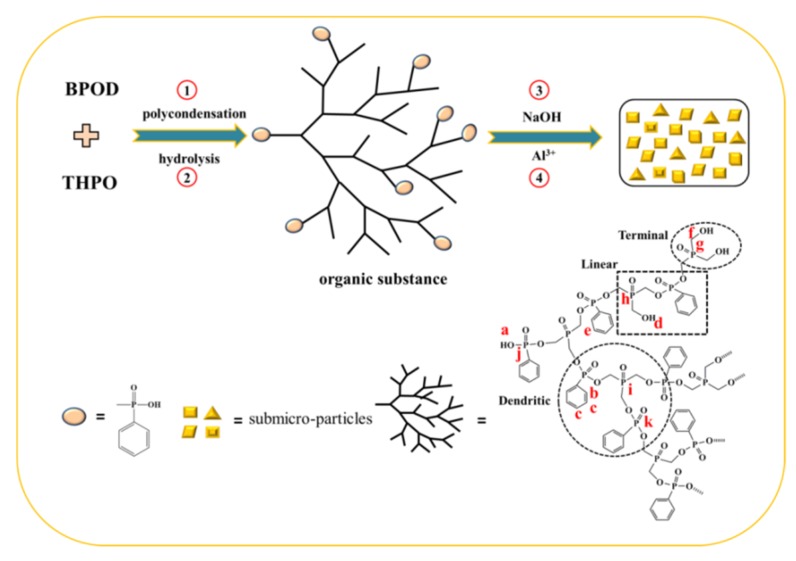
Synthetic route of hyper-branched aluminum phosphonate (AHPP) via the dehydrochlorination of tris(hydroxymethyl)phosphine oxide (THPO) with benzene phosphorus oxydichloride (BPOD).

**Figure 2 polymers-12-00064-f002:**
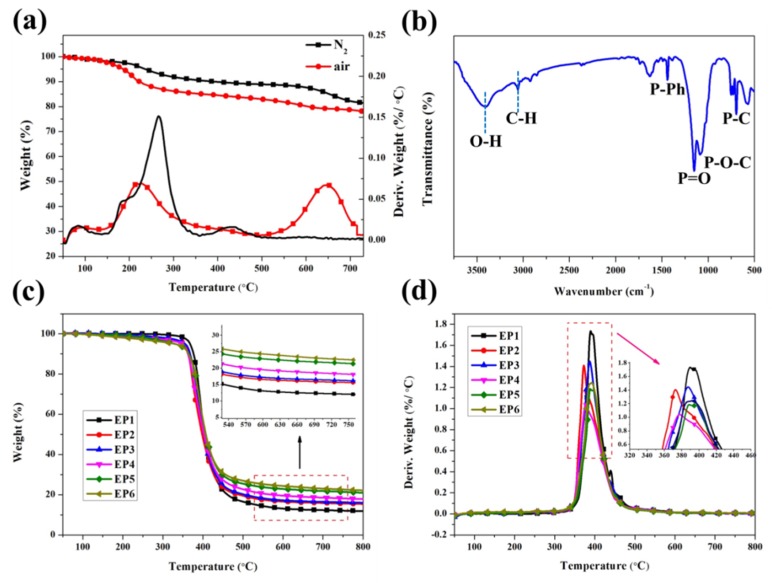
TGA and DTG curve of AHPP under nitrogen and air (**a**), FTIR spectrum (**b**) of AHPP, TGA (**c**) and DTG curves (**d**) of pristine EP and its composites under nitrogen.

**Figure 3 polymers-12-00064-f003:**
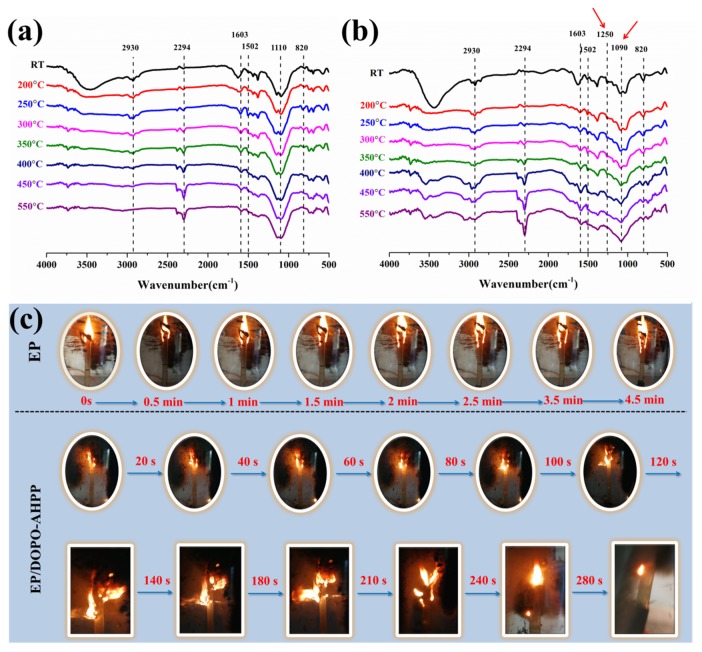
FTIR spectra of pristine EP (**a**), EP/DOPO-AHPP (**b**) at different pyrolysis temperatures, and video screenshots of pristine EP, EP/DOPO-AHPP at oxygen concentrations of 23.5% and 32% during LOI test (**c**).

**Figure 4 polymers-12-00064-f004:**
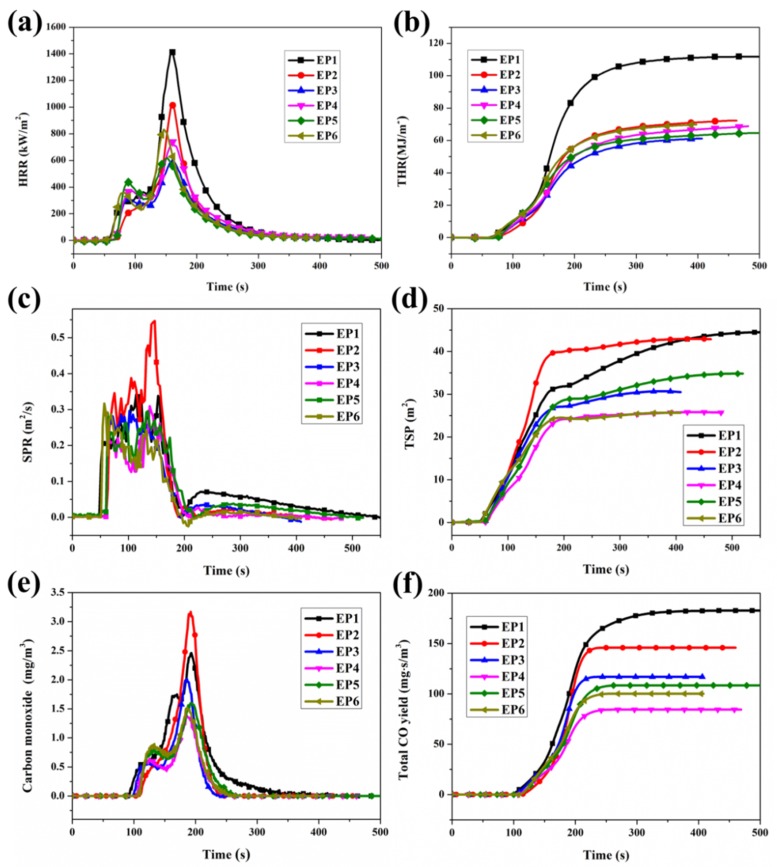
Heat release rate (**a**), total heat release (**b**) smoke production rate (SPR) (**c**), total smoke production (TSP) (**d**), CO (**e**), and total CO yield (**f**) versus temperature curves of pristine EP and its composites from cone test.

**Figure 5 polymers-12-00064-f005:**
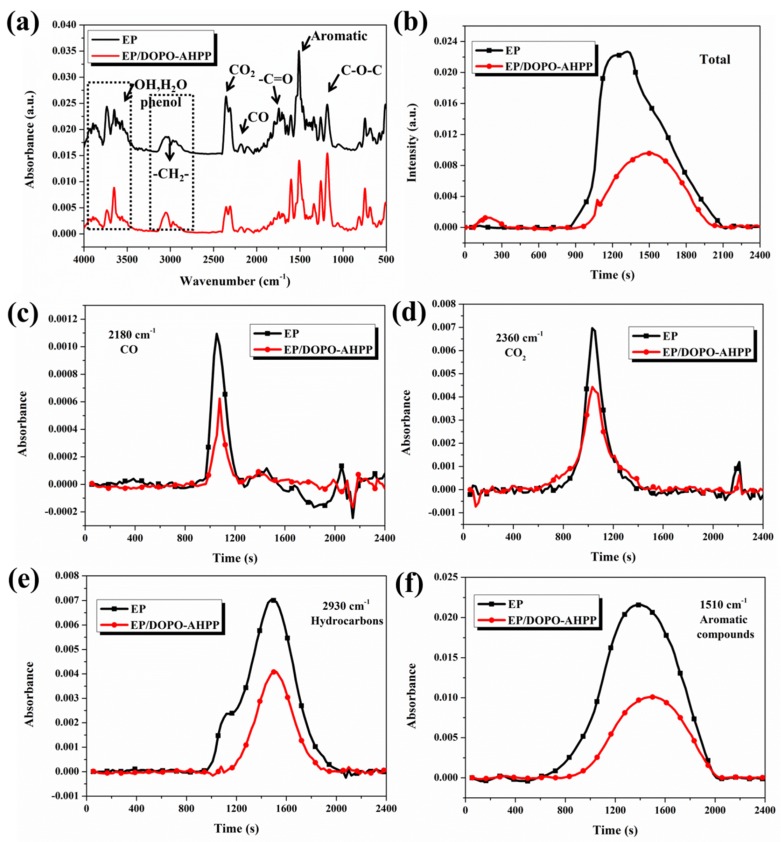
FTIR spectrum of pyrolysis products at the maximum decomposition rate (**a**); Gram–Schmidt curves (**b**) and absorbance of pyrolysis products in nitrogen atmosphere for EP and EP-4 versus time. CO (**c**); CO_2_ (**d**); hydrocarbons (**e**); and aromatic compounds (**f**).

**Figure 6 polymers-12-00064-f006:**
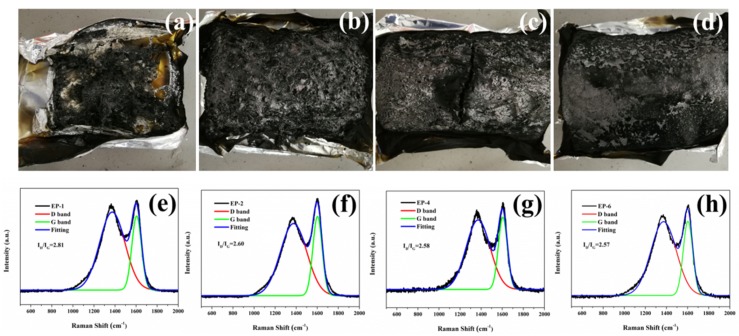
Photographs and Raman spectra of the char residue from pristine EP (**a**,**e**), EP/DOPO (**b**,**f**), EP/DOPO-AHPP (**c**,**g**), and EP-AHPP (**d**,**h**) after cone calorimeter tests.

**Figure 7 polymers-12-00064-f007:**
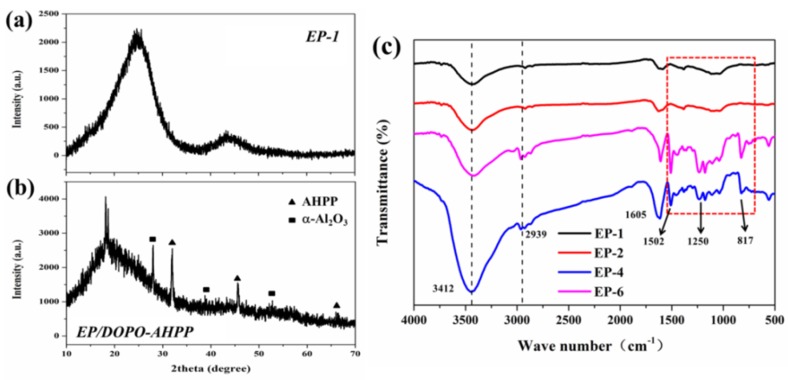
XRD patterns and FTIR spectra of the char residue of pristine EP and its composites from cone test.

**Figure 8 polymers-12-00064-f008:**
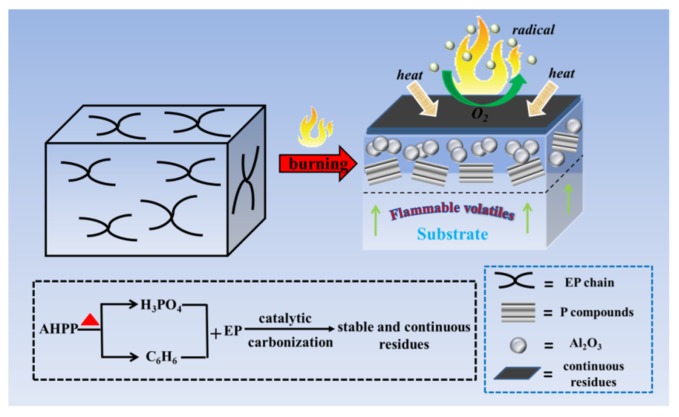
Illustration of the mechanism for the enhanced flame retardance and smoke toxicity suppression of EP/DOPO-AHPP composite.

**Table 1 polymers-12-00064-t001:** The formulations of pristine epoxy resin (EP) and its composites.

Samples	EP (g)	DDM (g)	DOPO (g)	AHPP (g)	Proportion (DOPO/AHPP)	Additive Loading (wt. %)
EP-1	100	21.8	-	-	-	-
EP-2	100	21.8	6.41	-	-	5
EP-3	100	21.8	4.27	2.13	2/1	5
EP-4	100	21.8	2.13	4.27	1/2	5
EP-5	100	21.8	3.21	3.21	1/1	5
EP-6	100	21.8	-	6.41	-	5

**Table 2 polymers-12-00064-t002:** TGA data for cured epoxy resin and its composites under N_2_ atmosphere.

Samples	Nitrogen		
	*T_d_* (°C)	*T_max_* (°C)	Char (%)
**EP-1**	367	391	11.8
EP-2	279	372	12.4
EP-3	356	388	15.6
EP-4	321	378	21.0
EP-5	350	390	17.5
EP-6	329	393	21.1

**Table 3 polymers-12-00064-t003:** LOI, UL-94 rating, and cone calorimeter data of cured epoxy resin and its composites.

Samples	LOI (%)	UL-94	*t*_1_/*t*_2_ (s/s)	PHRR (kW/m^2^)	THR (MJ/m^2^)	SPR (m^2^/s)	CO (mg/m^3^)	Residues (wt. %)
EP-1	23.5	NR	-	1422	111.9	0.343	2.43	4.3
EP-2	28.5	NR	-	1015	72.6	0.546	3.18	14.5
EP-3	32	V-0	2.4/1.7	578	60.9	0.295	1.59	18.4
EP-4	31	V-1	6.5/8.2	732	69.1	0.307	1.36	22.4
EP-5	31.5	V-0	1.5/4.2	607	65.0	0.316	1.52	21.0
EP-6	28	NR	-	832	69.4	0.294	2.00	18.9

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
