# Peer review of "Insight into Hyper-Branched Aluminum Phosphonate in Combination with Multiple Phosphorus Synergies for Fire-Safe Epoxy Resin Composites"

_polymers, 2020, doi:10.3390/polym12010064_

Round 1

Reviewer 1 Report

The manuscript entitled “Insight into hyper-branched aluminum phosphonate in combination with multiple phosphorus synergies for fire-safe epoxy resin composites” by Y. Yuan, B. Yu, W. Wang, Y. Shi, L. Mao, J. Xie, H. Pan, Y.Liu

The manuscript presents a method for improving the fire-resistant of epoxy resins, using a mixture of a hyper-branched aluminium phosphonate (AHPP) and 9,10-dihydro-9-oxa-10-phosphaphenanthrene-10-oxide (DOPO) as flame retardant. The preparation and characterization of AHPP and of AHPP-containing epoxy resins have already been presented in a previously published article (Journal of Hazardous Materials, 381, 121233, 2020). The novelty of the present manuscript consists in the preparation and characterization of epoxy resin composites containing two flame retardants DOPO (a commercial product) and AHPP (prepared by the authors). The introduction of both components, DOPO and AHPP, in epoxy resin was found to have a significant synergistic effect in improving the fire resistance of the resulting composites.

Some remarks:

The characteristics of DGEBA epoxy resin should be given (chemical structure, Epoxy Equivalent Weight). Which is the final structure of epoxy thermosets containing DOPO? Does DOPO react with reactive groups of epoxy resin? The authors should determine the glass transition of the samples and correlate it with the content of each flame retardant. If it is possible, the authors should measure the mechanical properties of the samples. Page 2, the hole name of TDBA abbreviation should be given. Page 3, it should be written “Tris(hydroxymethyl)phosphine oxide”, instead of “Trishydroxymethylphosphine oxide”. Page 6, the phrase “Accordingly, the result clearly indicates the formation of char residue for the EP composites with the simultaneous addition of AHPP and DOPO performs strong physical barriers during combustion to increase the fire safety of EP composites.” should be clarified.

The manuscript is well written, the characteristics of the new materials, especially the fire resistant properties, were investigated by using different techniques and the results were correctly interpreted. The fire resistant of epoxy resins was evaluated by measuring the Limiting oxygen index, performing UL-94 test, and determining the combustion behaviors by con calorimeter measurements. A potential flame retardation mechanism was also presented.

Therefore, I recommend the article for publication in “Polymers”.

Author Response

RE: Thank you for your kindly reminder and we are sorry for the inconsistent abbreviations. According to your kindly suggestions, the characteristics of DGEBA epoxy resin were added and the description was redrawn accordingly, which can be seen from page 2, 3 and 6, marked with blue color.

  The reactive P−H bond enables covalent binding of DOPO to the EP backbones or side chains by reaction with the epoxy functional group, which presents the EP with highly effective flame retardancy and stability with low probability of leaching from the polymer matrix. (Qiu, Y.; Liu, Z.; Qian, L.; Hao, J. Pyrolysis and flame retardant behavior of a novel compound with multiple phosphaphenanthrene groups in epoxy thermosets. J. Anal. Appl. Pyrolysis 2017, 127, 23-30). For the epoxy thermosets, with the increase of AHPP and DOPO content, the Tg values decreased. This phenomenon is ascribed to the plasticizing effect of flame retardant with lower Tg values, which enlarges the free volume of the epoxy resin composites. And the final structure of epoxy thermosets was shown in the uploaded attachment.

Reviewer 2 Report

 This work deals with flame retardation properties of epoxy resin composites bearing hyperbranched aluminum phosphonate and the conclusion was well derived from the experimental results. I think this manuscript can be acceptable after minor grammatical mistakes. Some English expressions contain typoes (e.g., AHPP vs. APHP in conclusion). 

Minors:

The Raman spectra (e-h) lost their captions in Figure 6. Specify 'e-h'. The authors make sure that all abbreviated terms should be fully noted before use.

Author Response

RE: Thanks a lot for your helpful suggestions and sorry for the missing part, and descriptions of the captions in Figure 6 and “APHP” were added and revised accordingly, which can be seen from page 10 and page 12, marked with blue color.
